# Factors Associated with Colorectal Polyps in Middle-Aged and Elderly Populations

**DOI:** 10.3390/ijerph19127543

**Published:** 2022-06-20

**Authors:** Song-Seng Loke, Seng-Kee Chuah

**Affiliations:** 1Division of Geriatric Medicine, Department of Family Medicine, Kaohsiung Chang Gung Memorial Hospital, Chang Gung University College of Medicine, 123, Dapi Road, Niaosong District, Kaohsiung 833, Taiwan; 2Division of Hepatogastroenterology, Department of Internal Medicine, Kaohsiung Chang Gung Memorial Hospital, Chang Gung University College of Medicine, 123, Dapi Road, Niaosong District, Kaohsiung 833, Taiwan; chuahsk@cgmh.org.tw

**Keywords:** colorectal polyps, low bone mineral density, metabolic syndrome, health examination

## Abstract

Colorectal polyps are the precursor lesions of most colorectal cancers. This study aimed to evaluate associations between bone mineral density (BMD), metabolic syndrome (MetS), and gastrointestinal diseases with colorectal polyps in middle-aged and elderly populations. A retrospective cross-sectional study was performed using data from the health examination database of a tertiary medical center in southern Taiwan in 2015. Subjects aged 50 years and older who had been assessed for metabolic factors and had undergone colonoscopy, upper gastrointestinal endoscopy, and dual energy X-ray absorptiometry scan (DEXA) were included. Factors associated with colorectal polyps were evaluated using univariate and multivariate logistic regression. In total, 1515 subjects were included, with mean age 60.1 years. Among them, 710 (46.9%) had colorectal polyps. Multivariate logistic regression analysis showed that high fasting glucose (OR = 1.08, *p* = 0.001), high triglycerides (OR = 1.02, *p* = 0.008), high total cholesterol (OR = 1.004, *p* = 0.009), reflux esophagitis (OR = 1.44, *p* = 0.002), duodenal polyps (OR = 1.75, *p* = 0.026), gastric ulcer (OR = 1.38, *p* = 0.024), duodenal ulcers (OR = 1.45, *p* = 0.028), osteopenia (OR = 1.48, *p* = 0.001), and MetS (OR = 1.46, *p* < 0.001) were independently associated with colorectal polyps. In conclusion, hyperglycemia, hypercholesterolemia, hypertriglyceridemia, MetS, duodenal polyps, gastric and duodenal ulcers, reflux esophagitis, and low BMD are independent risk factors associated with colorectal polyps in the middle-aged and elderly Taiwanese population.

## 1. Introduction

The incidence of colorectal cancer (CRC) has been increasing dramatically worldwide during the last two decades. CRC is the third most common cancer among men and the second most common cancer among women [1]. Colorectal polyps are precursor lesions of most colorectal cancers. Most colorectal polyps are asymptomatic and are typically ignored until malignancies have developed with symptoms of hematochezia and/or abdominal pain.

Several studies have shown that gender, age, body mass index (BMI), and metabolic syndrome (MetS) are associated with colorectal polyps [2,3,4,5,6]. Reflux esophagitis has also been found to be associated with colorectal polyps [7]; however, the relationships between other gastrointestinal diseases and colorectal polyps are not well studied. For individuals aged 50 years or older, the incidence of osteoporosis and colorectal polyps is increasing. Bone mineral density (BMD) is reported to be inversely associated with colorectal adenoma [8,9]. Since colorectal polyps and adenoma are precursor lesions for CRC, the correlation between colorectal polyps and BMD for populations aged 50 years and older has raised interest. The present study aimed to retrospectively evaluate the associations between colorectal polyps, BMD, and gastrointestinal diseases among subjects aged 50 years and older.

## 2. Materials and Methods

### 2.1. Study Design and Subjects

This retrospective cross-sectional study was performed using data extracted from the database of the Health Management and Evaluation Center of a tertiary medical center located in southern Taiwan from populations undergoing health examinations from January 2015 to December 2015. The center offers comprehensive medical tests and procedures as part of routine physical examinations. Among 5361 subjects aged from 20 to 87 years-old and 3120 (58.2%) subjects aged 50 years and over who underwent annual routine medical check-ups at the health examination center, most were free of symptoms. Subjects aged 50 years and older who had been assessed for metabolic factors and had undergone colonoscopy, upper gastrointestinal endoscopy, and dual energy X-ray absorptiometry scan (DEXA) were included. Subjects who were aged younger than 50 years; had not undergone colonoscopy, upper gastrointestinal endoscopy, or dual energy X-ray absorptiometry scan (DEXA); had incomplete data; or had prior or current colorectal cancer were excluded. Finally, the data of 1515 subjects were retained as the analytic sample. The study protocol was approved by the Institutional Review Board of the study hospital. (IRB No.: 202100126B0) on 25 February 2021. Because of the retrospective nature of the study, signed informed consent of subjects was waived.

### 2.2. Measurement of Anthropometric Parameters and Bone Mineral Density

Body height and weight were measured as the distance of the vertex point from the basis using the same testing equipment (HW-3030, Super-View Medical, Hualien, Taiwan) with accuracy of 1 mm and 0.1 kg, respectively. Subjects were standing upright, barefoot, and wearing light clothing. BMI was then calculated as weight in kilograms divided by height in meters squared (kg/m^2^). Waist circumference (WC) was measured in centimeters at the mid-level between the iliac crest and the lower border of the 12th rib while subjects stood with feet 25–30 cm apart. BMD values were measured in g/cm^2^ by DEXA (Lunar Prodigy Advance; GE Healthcare, Madison, WI, USA) at the lumbar spine, total femoral (total hip), and femoral neck.

### 2.3. Definition of Metabolic Syndrome and Lower Bone Mineral Density

In this study, MetS was defined according to the modified National Cholesterol Education Program Adult Treatment Panel III (NCEP ATP III) for Asian populations [10]. The modified NCEP ATP III criteria suggested the cut-off points of WC of Asian populations should use the cut-off of 90 cm in men and 80 cm in women. MetS was diagnosed when at least three of the following five components were found: (1) WC ≥ 90 cm for men and ≥ 80 cm for women; (2) high blood pressure (a systolic blood pressure ≥130 mm Hg and/or diastolic pressure ≥85 mm Hg, under treatment, or already diagnosed with hypertension); (3) high serum triglyceride (≥150 mg/dL); (4) decreased HDL-C (<40 mg/dL for males and <50 mg/dL for females); and (5) high fasting glucose (FG) ≥ 100 mg/dL, under treatment, or previously diagnosed with diabetes mellitus.

The diagnosis of osteoporosis was defined according to the World Health Organization (WHO) definition. The T-score was calculated automatically; the lowest value of the lumbar spine, total femoral (total hip), and femoral neck was chosen for the diagnosis of osteoporosis. Osteoporosis was defined as T-score ≤–2.5; osteopenia as −2.5 < T-score < −1; and normal as ≥−1. A definition of low BMD included both osteopenia and osteoporosis.

### 2.4. Statistical Analyses

Data are expressed as mean and standard deviation (mean ± SD) for continuous variables and frequency with counts and percentages (*n*, %) for categorical variables. The 𝜒^2^-test was used to compare subjects’ categorical characteristics, and the Student’s *t*-test was used to compare continuous variables. Multivariate logistic regression analysis was applied to determine odds ratios (ORs) and 95% confidence interval (CI). Pearson correlated coefficients were calculated between continuous data. To respect the assumption of independence between variables in a multivariable analysis, the highly correlated variable would not be included. Model 1 was conducted for all variables except parameters of MetS, and backward selection was used to establish the risk factors for colorectal polyps. Model 2 was conducted for all variables except MetS disease, and backward selection was used to establish the risk factors for colorectal polyps. Due to correlations between selected indicators of MetS, such as WC and HDL-C, stratified analysis was conducted for gender. A *p*-value < 0.05 was considered as statistically significant. SPSS software version 22.0 (IBM Corp, Armonk, NY, USA) was used for all statistical analysis.

## 3. Results

### 3.1. Prevalence of Colorectal Polyps and Lower BMD

In total, 1515 subjects were included in this study, with 624 (41.2%) females and 891 (58.8%) males. The mean age was 60.1 ± 6.6 years. The prevalence of colorectal polyps and low BMD were 46.9% and 47.7%, respectively.

### 3.2. Differences between the Subjects with and without Colorectal Polyps

Table 1 compares the characteristics between subjects with and without colorectal polyps. The mean ages of subjects with colorectal polyps and without colorectal polyps were 60.3 ± 6.6 years and 59.8 ± 6.6 years, respectively. No significant differences in age, sex, BMI, WC, systolic BP, and diastolic BP were found between the two groups. Compared with subjects without colorectal polyps, those with colorectal polyps had significantly higher fasting blood glucose (*p* < 0.001), total cholesterol (*p* = 0.004) and triglycerides (*p* < 0.001), and lower high-density lipoprotein cholesterol (HDL-C) (*p* = 0.008). Subjects with colorectal polyps also had significantly higher incidence of MetS (*p* < 0.001), reflux esophagitis (*p* = 0.003), duodenal polyps (*p* = 0.014), gastric ulcers (*p* = 0.007), duodenal ulcers (*p* = 0.030), and low BMD (*p* < 0.001).

Table 2 compares the characteristics between the subjects with and without colorectal polyps stratified by gender. Males had a similar trend with all patients. However, in females, fasting glucose (*p* = 0.001), triglycerides (*p* = 0.002), and MetS (*p* = 0.010) remained significantly associated with colorectal polyps.

### 3.3. Multivariate Logistic Regression Analyses of the Associations between Colorectal Polyps and Risk Factors

Results of multivariate logistic regression analyses are shown in Table 3. Due to the high correlation between BMI and WC (r = 0.833), systolic BP, and diastolic BP (r = 0.731, data not shown), BMI and diastolic BP were not included in multivariate analysis. For all patients, after backward selection, multivariate logistic regression analyses show that high total cholesterol (OR = 1.004, *p* = 0.009), reflux esophagitis (OR = 1.44, *p* = 0.002), duodenal polyps (OR =1.75, *p* = 0.026), gastric ulcers (OR =1.38, *p* = 0.024), duodenal ulcers (OR = 1.45, *p* = 0.028), osteopenia (OR = 1.48, *p* = 0.001), and MetS (OR = 1.46, *p* < 0.001) were independently and significantly associated with colorectal polyps (Model 1). Among the components of MetS, only high fasting glucose (OR = 1.008, *p* < 0.001) and high triglycerides (OR = 1.003, *p* < 0.001) were independently and significantly associated with colorectal polys (Model 2).

For males, after backward selection, multivariate logistic regression analyses showed that high total cholesterol (OR = 1.004, *p* = 0.031), reflux esophagitis (OR = 1.57, *p* = 0.003), duodenal ulcers (OR = 1.78, *p* = 0.009), osteopenia (OR = 1.70, *p* < 0.001), and MetS (OR = 1.49, *p* = 0.005) were independently and significantly associated with colorectal polyps (Male—Model 1). For females, after backward selection, multivariate logistic regression analyses showed that only MetS (OR = 1.51, *p* = 0.010) was independently and significantly associated with colorectal polyps (Female—Model 1).

Among the components of MetS, in both males and females, high fasting glucose and high triglycerides were independently and significantly associated with colorectal polys (Male/Female—Model 2).

## 4. Discussion

In the present study, the prevalence of colorectal polyps was 46.9% among subjects aged 50 years and older. In previous studies, the prevalence of colorectal polyps ranged from 30–50% [11,12,13,14,15,16], and the highest reported prevalence of polyps in a screening population was 58% [12]. The prevalence of colorectal polyps was higher in men than in women [13,14] and increased with age [15]. The prevalence of colorectal polyps in men and women was comparable to these findings: 46% vs. 48% in the present study. Data from the population-based National Health and Nutrition Examination Survey (NHANES) in 2017–2018 showed that the age-adjusted prevalence of osteoporosis among adults aged 50 and over was 12.6%, and the prevalence of low bone mass among adults aged 50 and over was 43.1% [16]. The prevalence of low BMD in the present study was 47.7%, with 37.2% classified as osteopenia and 10.5% classified as osteoporosis.

The present study showed that MetS, high fasting blood glucose, and hypertriglyceridemia were independent risk factors for colorectal polyps, consistent with previous findings. A systematic review by Passarelli et al. [17] showed that hypertriglyceridemia is associated with colorectal polyps. Xie et al. [18] showed high LDL and triglycerides levels are correlated with polyps. Studies also showed that type 2 diabetes mellitus (T2DM) increases the risk of colorectal polyps [19] and colorectal cancer [20]. In insulin-resistant T2DM patients, hyperinsulinemia and free IGF-1 may promote the proliferation of colonic epithelial cells, possibly contributing to development of polyps and tumorigenic effect [21,22,23].

The present study found that hypercholesterolemia was independently associated with colorectal polyps. However, the relationship between total cholesterol and colorectal polyps remains conflicting. Demers et al. [24] showed no association between total cholesterol and colorectal polyps after adjusting for age, while other studies found that cholesterol was positively associated with colorectal polyps [25,26]. However, cell proliferation enhanced by cholesterol as well as by triglycerides, as mentioned above, may be the causative mechanism for developing colorectal polyps [25].

Associations between colorectal polyps, gastrointestinal ulcers and polyps have not been studied previously. The present study showed that gastric and duodenal ulcers are independently associated with colorectal polyps. The mechanism of the association is not completely understood, but several studies have shown that colorectal polyps and colorectal cancer are associated with *Helicobacter pylori* (*H. Pylori*) infection [27,28,29,30,31,32]. Since *H*. *pylori* is also a risk factor for development of gastric and duodenal ulcers, both ulcers may be associated with colorectal polyps. The present study also found associations between gastric and duodenal polyps with colorectal polyps in the population without familial adenomatous polyposis. Studies have shown that patients with duodenal adenomas have higher prevalence of colonic adenomas compared to the general population [33,34]. Duodenal adenomas polyps are extremely common in patients with familial adenomatous polyposis. However, in patients without familial adenomatous polyposis, the frequency of colorectal adenomatous polyps also increased in patients with duodenal polyps compared with the general population [34]. The underlying mechanism still needs further clarification.

The present study also found that reflux esophagitis was associated with colorectal polyps. Several studies have found that Barrett’s esophagitis and erosive esophagitis are associated with colorectal polyps [7,35,36,37,38]. Since reflux esophagitis sometimes progresses to Barrett’s esophagitis when not effectively treated, it is reasonable that reflux esophagitis may also be associated with colorectal polyps.

The present study found an association between low BMD and colorectal polyps. Lim et al. [7] showed that osteoporosis was associated with the risk of colorectal adenoma in women, and Nock et al. [8] showed that low BMD was associated with an increased risk of colorectal adenoma. Calcium and vitamin D are necessary for the formation of bone mass, which is measured by BMD. Higher circulating levels of vitamin D metabolites and dietary intakes of calcium are inversely associated with the risk of colorectal adenomas [39,40,41]. Higher calcium levels have been shown to exert an anti-neoplastic effect in the colon epithelium by inducing higher levels of apoptosis [42,43]. Vitamin D also helps to maintain calcium homeostasis and may act directly on the colon epithelium by regulating apoptosis and cellular differentiation and by modulating growth factor and cytokine levels [44]. Given that low BMD implies loss of calcium homeostasis, the consequent effect may be associated with colorectal polyps.

The present study showed that gender differences were displayed in patients with low BMD and gastrointestinal diseases but not in those with MetS. Meanwhile, no associations were found in the present study between colorectal polyps and gender, age, and BMI, which may be illustrated by characteristics of the study population. It has been reported that age over 50–65 years and BMI higher than 25 are independent risk factors for colorectal polyps [2,6]; however, the present study was conducted in a population aged 50 years and older, and they had a mean BMI of 24.4. Possibly, the effects of age and BMI were negligible in our population. Besides, Wang et al. [6] also found that gender is not associated with adenomatous colorectal polyps in data from healthy examination database in south Taiwan.

This study has several limitations. First, the data were obtained from an administrative computer database and were analyzed in a study with retrospective, cross-sectional design, which limits inferences of causality such as the cause-and-effect of colorectal polyps and its associations with BMD and other variables that still need verification through further prospective study. Second, the study participants were from a single health-promotion center in Taiwan and may not be representative of the general population or other ethnic populations, which limits generalization to other populations. Third, lifestyle factors, such as smoking, alcohol consumption, and dietary habits, and physical activity that may affect the formation of colorectal polyps and BMD were not available in the database and could not be adjusted in statistical models. In addition, because of using retrospective data, pathology reports of polyps were not available for most subjects to complete our analysis. Lastly, further evaluation is needed to explore the possibility of two-way dependencies between colorectal polyps and its associated factors, such as how colorectal polyps can stimulate bone loss, while low bone mass may increase the risk of colorectal polyps.

## 5. Conclusions

Hyperglycemia, hypercholesterolemia, hypertriglyceridemia, MetS, reflux esophagitis, duodenal ulcers, duodenal polyps, and low BMD are independent risk factors associated with colorectal polyps. Further large prospective cohort studies are needed to make a more convincing case for these associations.

## Figures and Tables

**Table 1 ijerph-19-07543-t001:** Differences between colorectal polyps group and non-colorectal polyps group.

	Colorectal Polyps (*n* = 710)	Non-Colorectal Polyps (*n* = 805)	# *p*-Value
Age, years	60.3 ± 6.6	59.8 ± 6.6	0.121
Sex			0.429
Female, *n* (%)	300 (42.3)	324 (40.2)	
Male, *n* (%)	410 (57.7)	481 (59.8)	
BMI, kg/m2	24.4 ± 3.5	24.4 ± 3.5	0.886
WC, cm	83.4 ±10.1	83.5 ±10.1	0.807
Blood pressure (BP), mmHg			
Systolic	129.5 ± 19.6	128.7 ± 19.0	0.410
Diastolic	85.1 ± 11.3	85.0 ± 10.9	0.867
Fasting glucose, mg/dL	108.9 ± 28.6	102.1 ± 22.4	<0.001 *
Total cholesterol, mg/dL	211.2 ± 39.2	205.3 ± 39.8	0.004 *
HDL-C, mg/dL	47.3 ± 11.3	48.9 ± 12.3	0.008 *
Triglycerides, mg/dL	150.3 ± 107.7	126.5 ± 73.2	<0.001 *
LDL-C, mg/dL	134.7 ± 36.4	131.3 ± 36.4	0.070
MetS, *n* (%)	330 (46.5)	294 (36.5)	<0.001 *
Reflux esophagitis, *n* (%)	220 (31.0)	194 (24.1)	0.003 *
Gastric polyps, *n* (%)	113 (15.9)	103 (12.8)	0.083
Duodenal polyps, *n* (%)	45 (6.3)	29 (3.6)	0.014 *
Gastric ulcer, *n* (%)	136 (19.2)	113 (14.0)	0.007 *
Duodenal ulcer, *n* (%)	92 (13.0)	76 (9.4)	0.030 *
BMD, *n* (%)			<0.001 *
Normal	334 (47.0)	459 (57.0)	
Osteopenia	296 (41.7)	267 (33.2)	
Osteoporosis	80 (11.3)	79 (9.8)	

#: Categorical variables were compared by χ^2^ test, and continuous variables were analyzed by Student’s *t*-test; * indicates a significant difference, *p* < 0.05. BMI, body mass index; BMD, bone mineral density; BP, blood pressure; HDL-C, high-density lipoprotein cholesterol; LDL-C, low-density lipoprotein cholesterol; MetS, metabolic syndrome; WC, waist circumference.

**Table 2 ijerph-19-07543-t002:** Differences between colorectal polyps group and non-colorectal polyps group stratified by gender.

	Male			Female		
	Colorectal Polyps (*n* = 410)	Non-Colorectal Polyps (*n* = 481)	# *p*-Value	Colorectal Polyps (*n* = 300)	Non-Colorectal Polyps (*n* = 324)	# *p*-Value
Age, years	60.5 ± 6.6	60 ± 6.6	0.310	60.1 ± 6.7	59.4 ± 6.5	0.212
BMI, kg/m^2^	25 ± 3.3	25 ± 3.2	0.916	23.6 ± 3.7	23.5 ± 3.6	0.563
WC, cm	87.5 ± 9	87.4 ± 8.9	0.941	77.9 ± 8.8	77.8 ± 8.9	0.894
Blood pressure (BP), mmHg						
Systolic	129.8 ± 17.7	129.2 ± 18.1	0.609	129.1 ± 22	128 ± 20.3	0.495
Diastolic	87.8 ± 10.1	87.1 ± 10.1	0.310	81.4 ± 11.7	81.9 ± 11.4	0.616
Fasting glucose, mg/dL	109.3 ± 27.9	102.4 ± 23.7	<0.001 *	108.4 ± 29.4	101.5 ± 20.3	0.001 *
Total cholesterol, mg/dL	211.5 ± 37.6	205.7 ± 39.7	0.026 *	210.7 ± 41.4	204.7 ± 40	0.064
HDL-C, mg/dL	47.1 ± 11.4	48.7 ± 12.4	0.047 *	47.6 ± 11.1	49.2 ± 12.2	0.082
Triglycerides, mg/dL	148.8 ± 93.4	126.6 ± 69.3	<0.001 *	152.3 ± 124.7	126.4 ± 78.7	0.002 *
LDL-C, mg/dL	135.5 ± 35.4	131.9 ± 36.7	0.144	133.6 ± 37.8	130.4 ± 36	0.268
MetS, *n* (%)	177 (43.2)	162 (33.7)	0.004 *	153 (37.3)	132 (27.4)	0.010 *
Reflux esophagitis, *n* (%)	138 (33.7)	121 (25.2)	0.005 *	82 (20)	73 (15.2)	0.165
Gastric polyps, *n* (%)	63 (15.4)	61 (12.7)	0.249	50 (12.2)	42 (8.7)	0.192
Duodenal polyps, *n* (%)	28 (6.8)	18 (3.7)	0.038 *	17 (4.1)	11 (2.3)	0.171
Gastric ulcer, *n* (%)	79 (19.3)	66 (13.7)	0.025 *	57 (13.9)	47 (9.8)	0.132
Duodenal ulcer, *n* (%)	56 (13.7)	42 (8.7)	0.019 *	36 (8.8)	34 (7.1)	0.551
BMD, *n* (%)			0.001 *			0.109
Normal	191 (46.6)	279 (58.0)		143 (34.9)	180 (37.4)	
Osteopenia	175 (42.7)	151 (31.4)		121 (29.5)	116 (24.1)	
Osteoporosis	44 (10.7)	51 (10.6)		36 (8.8)	28 (5.8)	

#: Categorical variables were compared by χ^2^ test, and continuous variables were analyzed by Student’s *t*-test; * indicates a significant difference, *p* < 0.05. BMI, body mass index; BMD, bone mineral density; BP, blood pressure; HDL-C, high-density lipoprotein cholesterol; LDL-C, low-density lipoprotein cholesterol; MetS, metabolic syndrome; WC, waist circumference.

**Table 3 ijerph-19-07543-t003:** Multiple logistic regression analyses of associations between variables and colorectal polyps #.

Variables	All				Male				Female			
	Model 1		Model 2		Model 1		Model 2		Model 1		Model 2	
	Adj OR (95% CI)	*p*-Value	Adj OR (95% CI)	*p*-Value	Adj OR (95% CI)	*p*-Value	Adj OR (95% CI)	*p*-value	Adj OR (95% CI)	*p*-Value	Adj OR (95% CI)	*p*-Value
Total cholesterol, mg/dL (continuous)	1.004 (1.001–1.006)	0.009			1.004 (1.0004–1.007)	0.031						
LDL-C, mg/dL (continuous)			1.003 (1–1.006)	0.043								
Reflux esophagitis (Yes vs. No)	1.44 (1.15–1.82)	0.002	1.37 (1.08–1.73)	0.009	1.57 (1.17–2.12)	0.003	1.47 (1.09–1.98)	0.012				
Gastric polyps (Yes vs. No)			1.36 (1.01–1.83)	0.043								
Duodenal polyps (Yes vs. No)	1.75 (1.07–2.85)	0.026										
Gastric ulcer (Yes vs. No)	1.38 (1.04–1.82)	0.024	1.33 (1.002–1.77)	0.048								
Duodenal ulcer (Yes vs. No)	1.45 (1.04–2.02)	0.028	1.45 (1.03–2.02)	0.031	1.78 (1.15–2.74)	0.009	1.79 (1.15–2.77)	0.009				
BMD (vs. Normal)		0.002		0.002		0.002		0.001				
Osteopenia	1.48 (1.19–1.85)	0.001	1.49 (1.19–1.86)	0.001	1.70 (1.27–2.27)	<0.001	1.74 (1.3–2.34)	<0.001				
Osteoporosis	1.32 (0.93–1.88)	0.117	1.31 (0.92–1.87)	0.138	1.17 (0.74–1.85)	0.495	1.17 (0.74–1.85)	0.512				
MetS (Yes vs. No)	1.46 (1.19–1.81)	<0.001	-----	-------	1.49 (1.13–1.97)	0.005	-----	-------	1.51 (1.10–2.08)	0.010	-----	-------
WC, cm (continuous)	-----	-------			-----	-------			-----	-------		
Systolic BP, mg Hg (continuous)	-----	-------			-----	-------			-----	-------		
Fasting glucose, mg/dL (continuous)	-----	-------	1.008 (1.004–1.013)	<0.001	-----	-------	1.008 (1.002–1.014)	0.005	-----	-------	1.010 (1.003–1.018)	0.006
Triglycerides, mg/dL (continuous)	-----	-------	1.003 (1.001–1.004)	<0.001	-----	-------	1.003 (1.001–1.005)	0.001	-----	-------	1.002 (1.0005–1.004)	0.014
HDL-C, mg/dL (continuous)	-----	-------			-----	-------			-----	-------		

Adj OR, adjusted odds ratio; BMI, body mass index; BMD, bone mineral density; BP, blood pressure; HDL-C, high-density lipoprotein cholesterol; LDL-C, low-density lipoprotein cholesterol; MetS, metabolic syndrome, WC, waist circumference. #: Model 1 was conducted for all variables except 5 parameters of MetS and Model 2 was conducted for all variables except MetS disease.

## Data Availability

The datasets generated and/or analyzed during the current study are available from the corresponding author on reasonable request.

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
