# Peer review of "Factors Associated with Colorectal Polyps in Middle-Aged and Elderly Populations"

_ijerph, 2022, doi:10.3390/ijerph19127543_

Round 1
Reviewer 1 Report
Dear Authors,
Thank you for the opportunity to review this article. The manuscript describes an important and interesting topic: comprehensive, multidimensional search for the determinants of colon polyps in middle-
aged and elderly populations. The prevalence of polyps usually affects more than 50% of the population and can have serious health effects.
In my opinion, manuscript requires major corrections, especially in terms of methodology description, description of results and theoretical chapters - introduction and discussion.
Comments and Suggestions for Authors:
1. Keywords should be different than the words in the title.
2. In my opinion, the title is poorly worded in relation to the content of the manuscript. The title suggests an emphasis on BMD, while there is only one additional independent rather than dependent
variable. I believe the title should reflect the content more.
3. The introduction did not show a potential relationship between BMD and the presence of intestinal polyps. The introduction does not indicate a potential mechanism that would prove the relationship
between BMD and CRC. Reading the introduction, it is not known why the authors decided to investigate such dependencies. The introduction requires a significant development in theoretical foundations.
4. In the Material and Methods section, provide the number and date when the study protocol was approved by the Institutional Review Board.
5. Please fill in the details of anthropometry, with what anthropometric equipment and in the elaboration of which methodology were the anthropometric features measured?
6. In the description of the statistical methods, the assumptions of the logistic regression model 1 and 2 should be described.
7. In the Material and methods chapter, please complete all the required citations of works (concerning anthropometry of the metabolic syndrome reference values, etc.)
8. I believe that the model of one-way logistic analysis is redundant and does not give reliable results (correlation of independent variables). I believe that these results should be removed from the
manuscript.
9. In my opinion, due to the correlation of selected indicators, eg HDL with gender, the analyzes should be performed in separate groups based on gender. Please consider this by the authors.
10. I believe that the discussion needs to be deepened. The authors analyze many independent factors determining the occurrence of polyps, while in the discussion there are far too few attempts to explain
potential relationships.
11. In addition, it is important to note that perhaps in some cases the dependency is bidirectional. For example, colon polyps can stimulate bone loss. Authors should definitely develop the thread of two-
way dependencies in separate paragraphs of discussion or mention it in a limitation study.
12. Technical notes: the manuscript requires many minor changes, corrections, text ordering, remove unnecessary spaces, square "2", complete the missing n (%) eg in tab. 1, make corrections in the
literature.
Thank you.
Author Response
Reviewer 1
- Keywords should be different than the words in the title.
Author’s Response
Thank you very much for your comment. The keywords have been updated, and the title also has been modified.
- In my opinion, the title is poorly worded in relation to the content of the manuscript. The title suggests an emphasis on BMD, while there is only one additional independent rather than dependent
Author’s Response
Thank you very much for your comment. We have modified the title to:
Factors associated with colorectal polyps in middle-aged and elderly populations
- The introduction did not show a potential relationship between BMD and the presence of intestinal polyps. The introduction does not indicate a potential mechanism that would prove the relationship between BMD and CRC. Reading the introduction, it is not known why the authors decided to investigate such dependencies. The introduction requires a significant development in theoretical foundations.
Author’s Response
Thank you very much for your comment. We have revised the text in Introduction section as below:
“Bone mineral density (BMD) is reported inversely associated with the presence of colorectal adenoma. Given that colorectal polyps and adenoma are precursor lesions of CRC, the correlation between colorectal polyps and BMD for populations aged 50 years and older raises interest”. Please see Line 42-45.
- In the Material and Methods section, provide the number and date when the study protocol was approved by the Institutional Review Board.
Author’s Response
Thank you very much for your suggestion. We have provided the number and date when the study protocol was approved by the Institutional Review Board in the Material and Methods section.
IRB No.: 202100126B0 and date of approval: February, 25, 2021.
- Please fill in the details of anthropometry, with what anthropometric equipment and in the elaboration of which methodology were the anthropometric features measured?
Author’s Response
Thank you very much for your comment. We have updated the description of anthropometry measurement. Please see Line 67-70.
- In the description of the statistical methods, the assumptions of the logistic regression model 1 and 2 should be described.
Author’s Response
Thank you very much for your comment. We added the description of the logistic regression model 1 and 2 in the section of “2.4. Statistical analyses” as below:
“Multivariate logistic regression analysis was applied to determine odds ratios (ORs) and 95% confidence interval (CI). Pearson correlated coefficients were calculated between continuous data. To violate the assumption of independence between variables in a multi-variable analysis, the highly correlation would not be included. Model 1 was conducted by all variables, except parameters of MetS and using backward selection to establish the risk factor for colorectal polyps. Model 2 was conducted by all variables, except MetS disease and using backward selection to establish the risk factor for colorectal polyps.” Please see Line 97-104.
- In the Material and methods chapter, please complete all the required citations of works (concerning anthropometry of the metabolic syndrome reference values, etc.)
Author’s Response
Thank you very much for your comment. We have updated Material and methods chapter with required information. Please see Line 66-92.
- I believe that the model of one-way logistic analysis is redundant and does not give reliable results (correlation of independent variables). I believe that these results should be removed from the manuscript.
Author’s Response
Thank you very much for your comment. We have removed the one-way logistic analysis from the results.
- In my opinion, due to the correlation of selected indicators, eg HDL with gender, the analyzes should be performed in separate groups based on gender. Please consider this by the authors.
Author’s Response
Thank you very much for your comment. Due to the correlation of selected indicator, such as HDL-C and waist circumference were difference in cut-off value with gender, we further conducted the stratified analysis on gender. Please see updated Table 2 and 3.
- I believe that the discussion needs to be deepened. The authors analyze many independent factors determining the occurrence of polyps, while in the discussion there are far too few attempts to explain potential relationships.
Author’s Response
Thank you very much for your comment. We have revised the whole Discussion section. Please see Line 185-241.
- In addition, it is important to note that perhaps in some cases the dependency is bidirectional. For example, colon polyps can stimulate bone loss. Authors should definitely develop the thread of two-way dependencies in separate paragraphs of discussion or mention it in a limitation study.
Author’s Response
Thank you very much for your comment. We have added the possibility of two-way dependencies between colorectal polyps and its related factors, for example, colorectal polyps can stimulate bone loss, while low bone mass might increase the risk of colorectal polyps, were not further evaluated in the limitation of the study. Please see Line 250-253.
- Technical notes: the manuscript requires many minor changes, corrections, text ordering, remove unnecessary spaces, square "2", complete the missing n (%) eg in tab. 1, make corrections in the literature.
Author’s Response
Thank you very much for your comment. We have carefully revised the whole manuscript.
Reviewer 2 Report
In that retrospective cross-sectional study from a large Taiwan database, the authors found association of BMD, metabolic syndrome and gastrointestinal diseases with colorectal polyps in 1515 subjects aged 50 years and older who had undergone colonoscopy in addition to upper gastrointestinal endoscopy, DXA and assessment of metabolic factors. In contrast to literature, they did not find association with age, sex and BMI. The strength of the study is the large number of subjects. However, there are many limitations, including the design of the study, the lack of information about some well-known risk factors and how multivariate analyses were carried out.
I have some questions and comments for the authors to consider.
Introduction
The authors should develop the rationale of analyzing the association of BMD with colorectal polyps.
Methods
Reasons for evaluating metabolic factors and bone density in addition to colonoscopy are not mentioned and could have distorted the results. Authors should indicate the proportion of subjects aged 50 and over who received the full assessment.
No information about colorectal cancer is provided. Was it an exclusion criteria?
Authors should specify what densitometer was used. Was it the same for all patients? What does “radius head” mean?
The sites used for the diagnosis of osteoporosis should be indicated.
The words "lower BMD" should be replaced with "low BMD" throughout the manuscript.
Results
Since the risk factors could be different in serrated versus adenomatous polyps, could the authors provide the information about the type of polyps and could they analyze them separately?
Table 1 and univariate logistic regression analysis are redundant.
The multivariate analysis is not clear enough. Why did the authors adjusted for age and BMI since they did not find differences in univariate analysis? The variables included in the model 2 should be specified. Since several variables are probably highly correlated (such as BMI and waist circumference…) their inclusion in the model violates the assumption of independence between variables in a multivariable analysis. Thus, authors should check for collinearity between variables included in the model.
Discussion
The reasons why the authors did not find associations with well-known rik factors such as ge sex and BMI should be discussed.
Another limitation of the study that should be added is the lack of information about physical activity
Minor comments
Table 2 information about duodenal ulcer seems to be inverted
Author Response
Reviewer 2
Introduction
The authors should develop the rationale of analyzing the association of BMD with colorectal polyps.
Author’s Response
Thank you very much for your comment. We have revised the text in Introduction section as below:
“Bone mineral density (BMD) is reported inversely associated with colorectal adenoma. Since colorectal polyps and adenoma are precursor lesions of CRC, the correlation between colorectal polyps and BMD for populations aged 50 years and older raises interest”. Please see Line 42-45.
Methods
Reasons for evaluating metabolic factors and bone density in addition to colonoscopy are not mentioned and could have distorted the results. Authors should indicate the proportion of subjects aged 50 and over who received the full assessment.
Author’s Response
Thank you very much for your comment. Our study population is subjects aged 50 and older. We have updated the inclusion criteria. Please see Line 57-62.
No information about colorectal cancer is provided. Was it an exclusion criteria?
Author’s Response
Thank you very much for your comment. We have added colorectal cancer in the exclusion criteria.
Authors should specify what densitometer was used. Was it the same for all patients? What does “radius head” mean?
Author’s Response
Thank you very much for your comment. We have filled the information of densitometer. The same densitometer was used for all subjects.
We have revised “Radius head” as “distal 1/3 of the radius”.
The sites used for the diagnosis of osteoporosis should be indicated.
Author’s Response
Thank you very much for your comment. The lowest value of the lumbar spine, total femoral (total hip), femoral neck and radius head distal 1/3 of the radius was used for the diagnosis of osteoporosis. Please see Line 88-89.
The words "lower BMD" should be replaced with "low BMD" throughout the manuscript.
Author’s Response
Thank you very much for your comment. We have replaced "lower BMD" with "low BMD" throughout the manuscript.
Results
Since the risk factors could be different in serrated versus adenomatous polyps, could the authors provide the information about the type of polyps and could they analyze them separately?
Author’s Response
Thank you very much for your comment. We are sorry that we failed to get the pathological reports of polyps from most of the subjects, so we cannot perform further analysis separately. We have mentioned this in the limitation of the study. Please see Line 253-254.
Table 1 and univariate logistic regression analysis are redundant.
Author’s Response
Thank you very much for your comment. We have removed the original Table 1 and univariate logistic regression analysis
The multivariate analysis is not clear enough. Why did the authors adjusted for age and BMI since they did not find differences in univariate analysis? The variables included in the model 2 should be specified. Since several variables are probably highly correlated (such as BMI and waist circumference…) their inclusion in the model violates the assumption of independence between variables in a multivariable analysis. Thus, authors should check for collinearity between variables included in the model.
Author’s Response
Thank you very much for your comment. We have updated the multivariate analysis and related description in Method and Results sections as follows.
2.4. Statistical analyses
“Model 1 was conducted by all variables, except parameters of MetS and using backward selection to establish the risk factor for colorectal polyps. Model 2 was conducted by all variables, except MetS disease and using backward selection to establish the risk factor for colorectal polyps”.
3.3. Multivariate logistic regression analyses of the associations between colorectal polyps and risk factors
“Due to the highly correlation between BMI and WC (r=0.833), SBP and DBP (r=0.731), BMI and DBP were not include in multivariate analysis.”
Please see Line 101-104, Line 144-146.
Discussion
The reasons why the authors did not find associations with well-known risk factors such as age, sex and BMI should be discussed.
Author’s Response
Thank you very much for your comment. We have added a text in Discussion section. Please see Line 233-241.
Another limitation of the study that should be added is the lack of information about physical activity
Author’s Response
Thank you very much for your comment. We have added physical activity in the limitation section. Please see Line 248.
Minor comments
Table 2 information about duodenal ulcer seems to be inverted
Author’s Response
Thank you very much for your comment. We have updated Table 2.

Round 2
Reviewer 1 Report
Dear Authors,
Thank you very much for the opportunity to review this paper.
Thank you for taking into account my comments.
I suggest small changes in the text. In my opinion, the description of somatic measurement methods is still not satisfactory.
Please see the exemplary descriptions of anthropometric measurements.
https://pubmed.ncbi.nlm.nih.gov/25395402/
https://pubmed.ncbi.nlm.nih.gov/33253200/
Thank you.
Author Response
Thank you very much for your comment. We have revised the description of somatic measurement according to your suggestion.

Reviewer 2 Report
The authors have satisfactorily answered to most of questions and made the necessary changes in the manuscript. However, the following concerns still need attention:
According to literature, the radius site should not be used for defining low bone mass
In the statistical section “to violate..” should be replaced by “to respect” .
In the multivariate analyses, the authors should clarify how the variables cholesterol, LDL-C, HDL-C, fasting glucose and triglycerides have been considered (category high vs not high or continuous values?)
Finally, an editing of English language and style is required.
Author Response
The authors have satisfactorily answered to most of questions and made the necessary changes in the manuscript. However, the following concerns still need attention:
According to literature, the radius site should not be used for defining low bone mass
Author’s Response:
Thank you very much for your comment. We have removed it from the manuscript.
In the statistical section “to violate..” should be replaced by “to respect” .
Author’s Response:
Thank you very much for your comment. We have corrected it.
In the multivariate analyses, the authors should clarify how the variables cholesterol, LDL-C, HDL-C, fasting glucose and triglycerides have been considered (category high vs not high or continuous values?)
Author’s Response:
Thank you very much for your comment. We have updated the “variable” volume in Table 3 with the continuous variables using (continuous) and categorical data using (Yes vs No) or (vs Normal(ref)).
Finally, an editing of English language and style is required.
Author’s Response:
Thank you very much for your comment. The manuscript has been edited by native speaker.
